# Evaluating the fitness of PA/I38T-substituted influenza A viruses with reduced baloxavir susceptibility in a competitive mixtures ferret model

Leo Y Lee[1,¤], Jie Zhou[2], Paulina Koszalka[1,3], Rebecca Frise[2], Rubaiyea Farrukee[4], Keiko Baba[5], Shahjahan Miah[6], Takao Shishido[5], Monica Galiano[7], Takashi Hashimoto[5], Shinya Omoto[5], Takeki Uehara[5], Edin J. Mifsud[1,4], Neil Collinson[8], Klaus Kuhlbusch[9], Barry Clinch[8], Steffen Wildum[9], Wendy S. Barclay[2], Aeron C. Hurt[1,4,9]*

1 WHO Collaborating Centre for Reference and Research on Influenza, at the Peter Doherty Institute for Infection and Immunity, Melbourne, Australia, 2 Department of Infectious Disease, Imperial College London, London, United Kingdom, 3 Biomedicine Discovery Institute & Department of Microbiology, Monash University, Victoria, Australia, 4 Department of Microbiology and Immunology, University of Melbourne, at the Peter Doherty Institute for Infection and Immunity, Parkville, Australia, 5 Shionogi & Co., Ltd, Osaka, Japan, 6 Public Health England, London, United Kingdom, 7 Francis Crick Institute, London, United Kingdom, 8 Roche Products Ltd, Welwyn Garden City, United Kingdom, 9 F. Hoffmann-La Roche Ltd, Basel, Switzerland

¤ Current address: Department of Microbiology and Immunology, University of Melbourne, at the Peter Doherty Institute for Infection and Immunity, Parkville, Australia
* aeron.hurt@roche.com

**Data Availability Statement:** All relevant data are within the manuscript and its Supporting Information files.

## Abstract

Baloxavir is approved in several countries for the treatment of uncomplicated influenza in otherwise-healthy and high-risk patients. Treatment-emergent viruses with reduced susceptibility to baloxavir have been detected in clinical trials, but the likelihood of widespread occurrence depends on replication capacity and onward transmission. We evaluated the fitness of A/H3N2 and A/H1N1pdm09 viruses with the polymerase acidic (PA) I38T-variant conferring reduced susceptibility to baloxavir relative to wild-type (WT) viruses, using a competitive mixture ferret model, recombinant viruses and patient-derived virus isolates. The A/H3N2 PA/I38T virus showed a reduction in within-host fitness but comparable between-host fitness to the WT virus, while the A/H1N1pdm09 PA/I38T virus had broadly similar within-host fitness but substantially lower between-host fitness. Although PA/I38T viruses replicate and transmit between ferrets, our data suggest that viruses with this amino acid substitution have lower fitness relative to WT and this relative fitness cost was greater in A/H1N1pdm09 viruses than in A/H3N2 viruses.

## Author summary

Influenza viruses are associated with considerable disease burden and circulate annually causing seasonal epidemics. Antiviral drugs can be used to treat influenza infections and

**Funding:** This work was funded by F. Hoffmann-La Roche (https://www.roche.com/) and Shionogi & Co. Ltd (https://www.shionogi.com/eu/en/) who both played a role in the study design, data collection and analysis, decision to publish and preparation of the manuscript. The Melbourne WHO Collaborating Centre for Reference and Research on Influenza is supported by the Australian Government Department of Health, who had no role in study design, data collection and analysis, decision to publish, or preparation of the manuscript.

**Competing interests:** I have read the journal's policy and the authors of this manuscript have the following competing interests: • Keiko Baba, Takashi Hashimoto, Shinya Omoto, Takao Shishido and Takeki Uehara are employees of Shionogi & Co. • Klaus Kuhlbusch, Steffen Wildum and Aeron C Hurt are employees of F. Hoffmann-La Roche. • Neil Collinson and Barry Clinch are employees of Roche Products Ltd. • Wendy Barclay has received honoraria from Roche, Sanofi Pasteur and Seqirus. • Leo YY Lee has received honoraria from Roche. • Paulina Koszalka, Jie Zhou, Rubaiyea Farrukee, Rebecca Frise, Edin Mifsud, Monica Galiano and Shahjahan Miah have nothing to disclose.

help reduce the disease burden. Occasionally, treatment can lead to the emergence of viruses with reduced antiviral susceptibility. Normally such viruses have reduced 'fitness', meaning they do not tend to spread or transmit widely, however on rare occasions, oseltamivir-resistant variants have become widespread in the community, thereby reducing the utility of the drug for treatment. Baloxavir is an antiviral recently licensed in many parts of the world for the treatment of influenza. Viruses with reduced susceptibility to baloxavir have been observed in clinical trials, but the frequency of such variants in the community has remained low (<0.1% globally since 2017–2018). We evaluated the fitness of viruses in ferrets and found that although A/H1N1 and A/H3N2 viruses with reduced baloxavir susceptibility were able to replicate and transmit among ferrets, they had a moderate reduction in fitness compared to normal 'wild-type' viruses, suggesting a reduced likelihood of spread. Surveillance to monitor for the frequency of viruses with reduced baloxavir susceptibility remains important.

## Introduction

Antivirals are important for the treatment of influenza infections, particularly in high-risk individuals such as the immunocompromised and the elderly. Neuraminidase inhibitors (NAIs) such as oseltamivir are the current standard of care for treating influenza-related hospitalizations during seasonal epidemics [1], and are stockpiled in some countries as pandemic contingency [2]. However, the high frequency of mutations during influenza virus replication combined with the selective pressure of antiviral treatment can lead to the emergence of viral variants with reduced antiviral susceptibility or resistance. The World Health Organization Global Influenza Surveillance and Response System regularly monitors circulating influenza viruses for antiviral resistance, as the prevalence of resistant variants circulating in the community is a key factor when considering which influenza antiviral drugs should be prescribed for clinical use. For example, M2 ion channel blockers are no longer prescribed to treat influenza A viruses, as nearly 100% of circulating influenza viruses contain an S31N amino acid substitution that confers resistance to these antiviral drugs [3]. Similarly, H275Y substitutions in viral neuraminidase (NA) can lead to reduced susceptibility of influenza viruses to oseltamivir and another NAI, peramivir [4]. While localized clusters of NA/H275Y oseltamivir-resistant A/H1N1pdm09 viruses have been detected [5,6], these variants are currently only detected at a frequency of <1% among circulating viruses [7]. There is an ongoing need for the development of antivirals that utilize novel mechanisms of action, such that multiple effective treatment options are available in the case that resistance towards an existing class of antiviral drugs becomes widespread.

Baloxavir (pro-drug: baloxavir marboxil; active form: baloxavir acid) has been approved for the treatment of uncomplicated influenza in otherwise-healthy and high-risk patients in several countries around the world. The influenza polymerase complex is a heterotrimer comprising the polymerase acidic (PA) and two polymerase basic (PB1 and PB2) proteins. Baloxavir targets the highly conserved cap-dependent endonuclease region of the PA to inhibit 'cap-snatching', a critical stage of influenza virus replication [8]. Baloxavir has activity against influenza A, B, C and D viruses [9,10].

Human clinical trial data have shown that a single oral baloxavir dose alleviates influenza symptoms at least as effectively as a five-day oral course of twice-daily oseltamivir, as well as suppressing viral shedding more rapidly in all patient populations examined [11–14]. In 9.7% of otherwise-healthy patients treated with baloxavir in the phase 3 CAPSTONE-1 trial, viruses

with amino acid substitutions at residue I38 of the PA protein were identified. The most frequent variant was PA/I38T, which has subsequently been shown to exhibit reduced susceptibility to baloxavir *in vitro* [11]. PA/I38T variants have emerged following baloxavir treatment in subsequent clinical trials, with the highest rate of 23.4% observed in pediatric patients [8,9,12]. Although PA/I38X variants are associated with up to 50-fold reduction in baloxavir susceptibility *in vitro* compared to wild type (WT) viruses [15], clinical efficacy (time to alleviation of symptoms) is still observed in baloxavir-treated patients in whom these variants develop, presumably in part because resistance does not emerge until several days after treatment has begun [16]. Although based on small patient numbers, post-hoc analyses of the phase 3 CAPSTONE-1 trial suggested these variants can be associated with prolonged virus detection and uncommonly with symptom rebound [16]. However, analysis of the CAPSTONE-2 trial (patients at high risk of complications) showed the opposite effect, whereby PA/I38X variants were associated with numerically faster resolution of symptoms than those without [14], suggesting further investigation into the clinical impact of PA/I38X variants is required.

The public health risk posed by variant viruses with reduced baloxavir susceptibility depends on their capacity to replicate and transmit compared with WT viruses, and their consequent prevalence in the community. Recent surveillance data show that the circulation of influenza virus variants harboring PA amino acid substitutions associated with reduced baloxavir susceptibility remains low, at <0.1% globally since 2017–2018 [7,17]. As previously observed in NAI-resistant viruses, amino acid substitutions that reduce influenza antiviral susceptibility can also lower the efficiency of viral replication and transmission [18–21]. The extent of these fitness impacts can vary depending on the specific substitution and the viral genetic background, and the development of compensatory mutations that may restore viral fitness in resistant variants [4,22–24].

The goal of our study was to evaluate the intrinsic fitness of PA/I38T-variant A/H1N1pdm09 and A/H3N2 viruses to inform their risk for community circulation. We used *in vitro* assays and *in vivo* ferret models to compare their replication and transmission efficiency relative to baloxavir-sensitive (WT) viruses, using both recombinant viruses and patient-derived clinical virus isolates from human trials.

## Results

### Baloxavir susceptibility of PA/I38T-variants from post-treatment clinical specimens

Pre- and post-treatment clinical isolate influenza A viruses were tested by plaque reduction assay to assess the baloxavir susceptibility of treatment-emergent variant viruses harboring the PA/I38T substitution. The mean baloxavir $EC_{50}$ of viruses from pre-treatment samples (WT) were 1.1–1.5 and 0.31–0.39 ng/mL in A/H1N1pdm09 and A/H3N2 viruses, respectively, whereas the corresponding baloxavir $EC_{50}$ of the post-treatment viruses (PA/I38T) were 82–87 and 36–49 ng/mL, respectively. The type B clinical isolate pair was plaque-picked from post-baloxavir treatment samples and the mean baloxavir $EC_{50}$ values for WT and PA/I38T viruses were 4.0 and 18 ng/mL, respectively. Therefore, A/H1N1pdm09, A/H3N2, and type B PA/I38T-variants displayed 58–77-, 93–155-, and 4.6-fold reductions in baloxavir susceptibility compared with their respective WT viruses, similar to levels reported previously [9,16,25].

### *In vitro* comparison of WT and PA/I38T-variant replication kinetics

The *in vitro* replicative capacity of pure populations of WT and PA/I38T-substituted clinical isolates was compared in MucilAir human nasal epithelial cells (Fig 1A). For the two A/

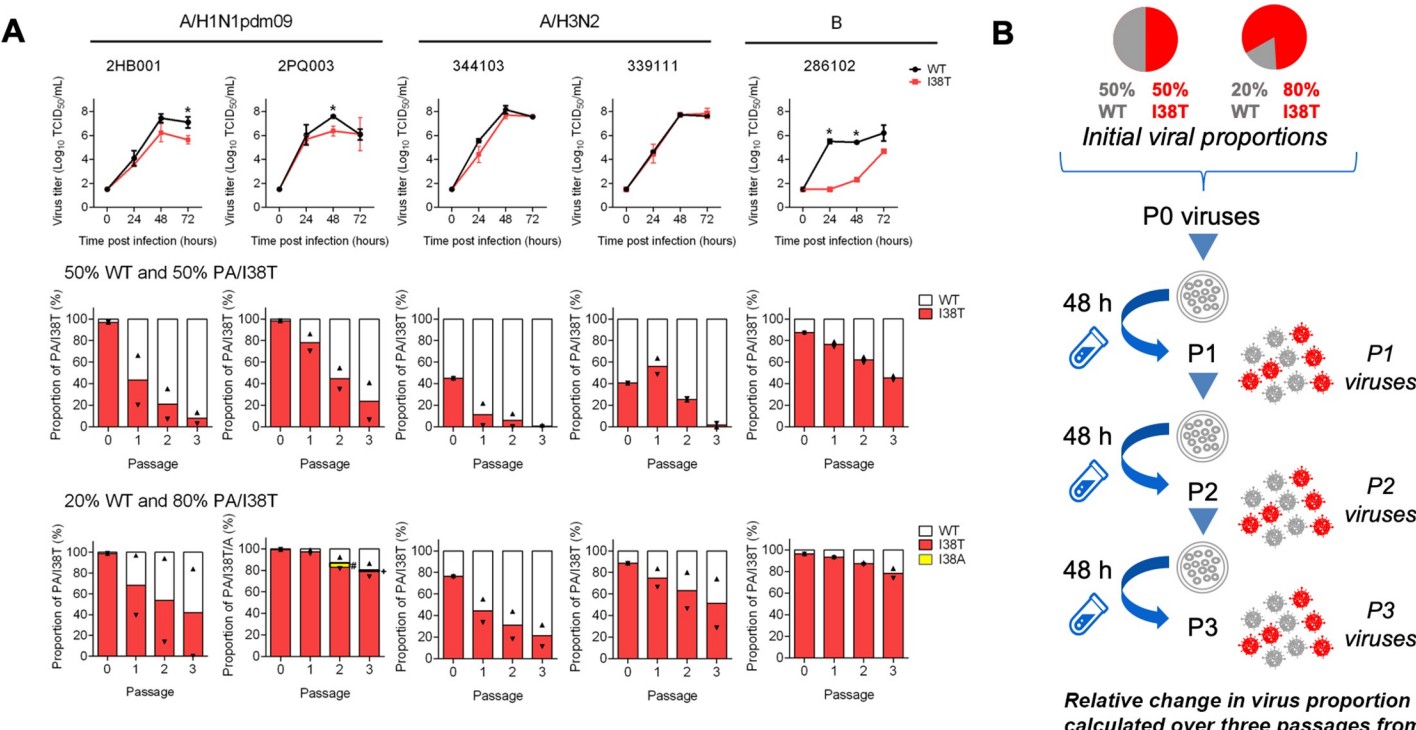

**Fig 1. Replicative capacity and competitive fitness of influenza PA/I38T substituted A/H1N1pdm09, A/H3N2 and B viruses isolated from clinical samples.** (A) MucilAir human nasal epithelial cells were infected with influenza clinical isolates at an MOI of 0.001 and 0.01 TCID$_{50}$/cell for type A and type B viruses, respectively. Culture fluids were collected at 0, 24, 48 and 72 hours post-infection and viral titers were determined in MDCK-SIAT1 cells. The LLOQ for virus titer was 1.57 log$_{10}$ TCID$_{50}$/mL and the viral titer lower than LLOQ was defined as 1.50 log$_{10}$ TCID$_{50}$/mL. Statistical comparisons of virus titers between WT and viruses with PA/I38T at each time-point were conducted using Welch's $t$-test at the 0.05 significance level ($^*$p<0.05). Data represent mean ± SD of triplicate experiments. MucilAir cells were co-infected with WT and PA/I38T-substituted clinical isolates at 50:50 or 20:80 ratio based on viral titers (TCID$_{50}$) at an MOI of 0.002 and 0.02 for type A and type B viruses, respectively. At 48 hours post-infection, the culture fluids were collected and serially passaged three times at an MOI of 0.001 and 0.01 for type A and type B viruses, respectively. Each culture fluid was subjected to NGS to determine the population of amino acids at position 38 in the PA subunit. The lower detection limit for calling variant viruses was 1% and variant population less than the lower detection limit was set as 0 for the calculation. The bar and the triangles indicate the mean and the individual values of duplicate experiments, respectively. The symbols (# and +) indicate 3.8% and 1.5% of PA/I38A proportion. (B) Experimental setup for the competitive mixtures experiments is depicted. LLOQ, lower limit of quantitation; MOI, multiplicity of infection; NGS, next generation sequencing; PA, polymerase acid; SD, standard deviation; TCID$_{50}$, 50% tissue culture infectious dose; WT, wild type.

H1N1pdm09 virus pairs, we observed transient reductions of viral titers in the PA/I38T viruses compared to WT at 48 hours in sample 2PQ003 (p<0.05) and at 72 hours (p<0.05) in sample 2HB001. No significant differences were observed in virus titers for A/H3N2 clinical isolates, indicating that *in vitro* growth kinetics of the A/H3N2 PA/I38T-variants were comparable to the WT viruses. Interestingly, we observed significantly lower viral titers (p<0.05) of PA/I38T-substituted influenza B isolates versus WT at 24 and 48 hours, suggesting the PA/I38T substitution reduces the intrinsic *in vitro* replication fitness of the influenza B clinical isolate (Fig 1A).

A competitive mixtures experiment design was then used to further investigate the effect of PA/I38T on *in vitro* replication fitness. Competitive mixtures experiments can be used to compare the relative difference in fitness between a variant and the corresponding WT virus by mixing both viruses in the initial inoculum at different ratios and examining the change in proportion for each virus population (Fig 1B). We evaluated variant viral fitness based on the change in the percentage of PA/I38T viruses (%I38T) in the viral population over three successive passages harvested from the infected MucilAir cell supernatants. For all influenza type A

and B virus pairs tested, the proportion of the PA/I38T virus replicating in MucilAir cells decreased with each passage, regardless of virus subtype or initial ratio (Fig 1A). Interestingly, a new PA/I38A variant virus was detected by next generation sequencing (NGS) in a #2PQ003 co-infection group at P2 and P3 (Fig 1A), however this subpopulation was transient and remained at low abundance. Compared to the A/H3N2 clinical isolates, the ratio of WT%:I38T % determined by NGS for A/H1N1pdm09 clinical isolate pairs did not correlate well with the intended mixture proportions based on infectious virus titer (used for preparing the inoculum [P0] material), which may be due to the presence of non-replicating virions present in the viral samples (Fig 1A). Despite these mixtures consisting of ~99% PA/I38T viral copies based on NGS, mixed virus populations similar to the expected proportions were established at 48 hours following inoculation (P1, [Fig 1A]), and we identified that the A/H1N1pdm09 PA/I38T-variant population declined compared to WT with each successive passage.

The *in vitro* replication fitness deficit observed in PA/I38T-variant clinical isolates was recapitulated in recombinant virus pairs. We performed competitive mixture experiments in MDCK cells using 50:50 ratios of recombinant WT and PA/I38X-variant virus pairs. A/H1N1 and A/H3N2 WT viruses outcompeted all of their respective PA/I38X variants tested over three passages *in vitro* (S1 and S2 Figs). Subsequently, we conducted identical competition experiments using recombinant WT/I38T virus pairs in the presence of baloxavir to understand fitness in the presence of different drug concentrations (S2 Fig). Whereas the WT virus population became fixed (near 100% purity by sequencing) at P3 in typical passaging conditions, PA/I38T variants became dominant at P3 for both A/H1N1 and A/H3N2 virus backgrounds in the presence of baloxavir at a concentration of 1x $EC_{50}$ [15]. The fixation of PA/I38T variants occurred earlier (by P1–2) when evaluated at higher concentrations of baloxavir (10–100x $EC_{50}$); however, in the presence of 100x $EC_{50}$, virus titer in the cell culture supernatant at P1 was 2.9 and 1.9 $\log_{10}$ median tissue culture infectious dose [$TCID_{50}$]/mL lower for rgA/WSN/33 (H1N1) and rgA/Victoria/3/75 (H3N2) viruses, respectively, compared with the condition in the absence of baloxavir (S2 Fig). When evaluated at 1000x $EC_{50}$ of baloxavir, viral replication of both WT and PA/I38T viruses were suppressed to such a degree that insufficient infectious virus was present to enable subsequent passage (S2 Fig).

To interrogate the relative intrinsic fitness of PA/I38T clinical isolates in a relevant *in vivo* model, the A/H3N2 clinical isolate pair #344103 (WT and PA/I38T) and A/H1N1pdm09 clinical isolate pair #2HB001 (WT and PA/I38T) were selected for evaluation of viral fitness in the ferret infection model (S3 Fig).

## Replication and direct contact transmission of WT/I38T clinical isolate virus pairs in a competitive fitness ferret model (Melbourne)

**A/H3N2 PA/I38T fitness.** In the Melbourne laboratory, the *in vivo* replication and transmission capacity of the post-treatment A/H3N2 PA/I38T isolate was compared to the corresponding pre-treatment WT isolate in ferret transmission chains (Fig 2A). Infectious virus was detected in nasal washes of each recipient ferret (RF) in transmission chains initiated with pure populations of either the WT or PA/I38T virus, indicating that the treatment-emergent PA/I38T-variant was capable of at least three successive transmissions in a direct contact model (Fig 3A). Pyrosequencing analysis showed that the PA/I38T-variant population was maintained over three generations of transmission without substantial reversion to WT (S5 Fig). To compare replication kinetics of A/H3N2 PA/I38T virus with WT virus *in vivo*, the area under the curve (AUC) of infectious viral titers in the nasal washes of RF1-RF3 (n = 3) was determined over the course of infection (Fig 3A). Similar to the *in vitro* experiments (Fig 1A), overall RF virus replication kinetics (AUC of the $TCID_{50}$ ± standard deviation [SD]) for

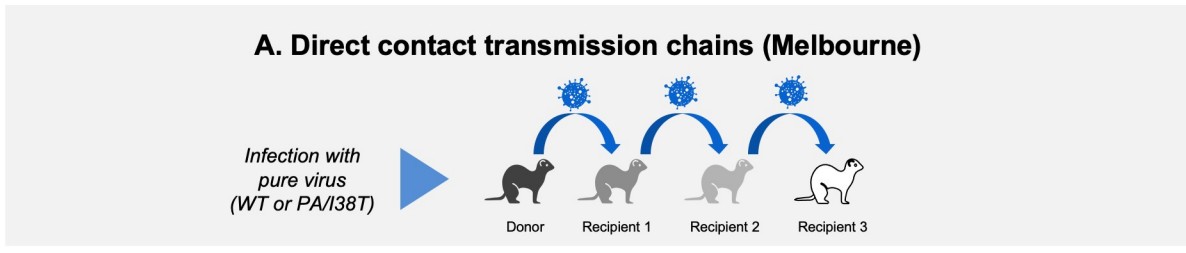

**Fig 2. Experimental setups used for the ferret transmission experiments. (A).** Direct contact transmission chains were used to examine fitness of pure WT or PA/I38T virus populations. (B and C) Donor ferrets are infected with different initial proportions of WT:I38T viruses in a direct contact model (B) or in both direct and indirect contact models (C). The relative change in virus proportions is calculated as indicated in the different models. WT, wild type.

the A/H3N2 WT virus (18.42 ± 7.29) were comparable to the PA/I38T virus (17.58 ± 2.75, non-significant).

In parallel, transmission chains were initiated with competitive virus mixtures in donor ferrets (DFs) inoculated with different ratios of A/H3N2 WT and PA/I38T viruses based on infectious titers (Fig 2B). Our aim was to establish infections in DFs with WT and PA/I38T-variant viruses present at a range of different ratios, allowing us to evaluate changes over time for the proportion of PA/I38T-variant population in infected ferrets and the differences in variant population following a transmission event to a recipient ferret. Pyrosequencing of viral RNA shedding in nasal wash samples showed that the mixed infections established in DFs at 1 day post inoculation (DPI) were similar to the intended infectious virus ratios in the inoculum mixture (Fig 4). To evaluate the relative within-host fitness of the PA/I38T-variant, in each individual ferret the %I38T population on the final day of nasal washing was compared to the population present on the first day of detection by pyrosequencing. We observed that the proportion of A/H3N2 PA/I38T remained relatively stable in DFs, with an overall change in PA/I38T over time of +7.9% for the (WT:I38T) 80:20 initial proportion, -2.8% and -6.9% for the 50:50 initial proportions, and -17.7% for the 20:80 initial proportion (Fig 4). In RF1s however, a decrease in the proportions of A/H3N2 PA/I38T was observed over time of -7.5% (80:20),

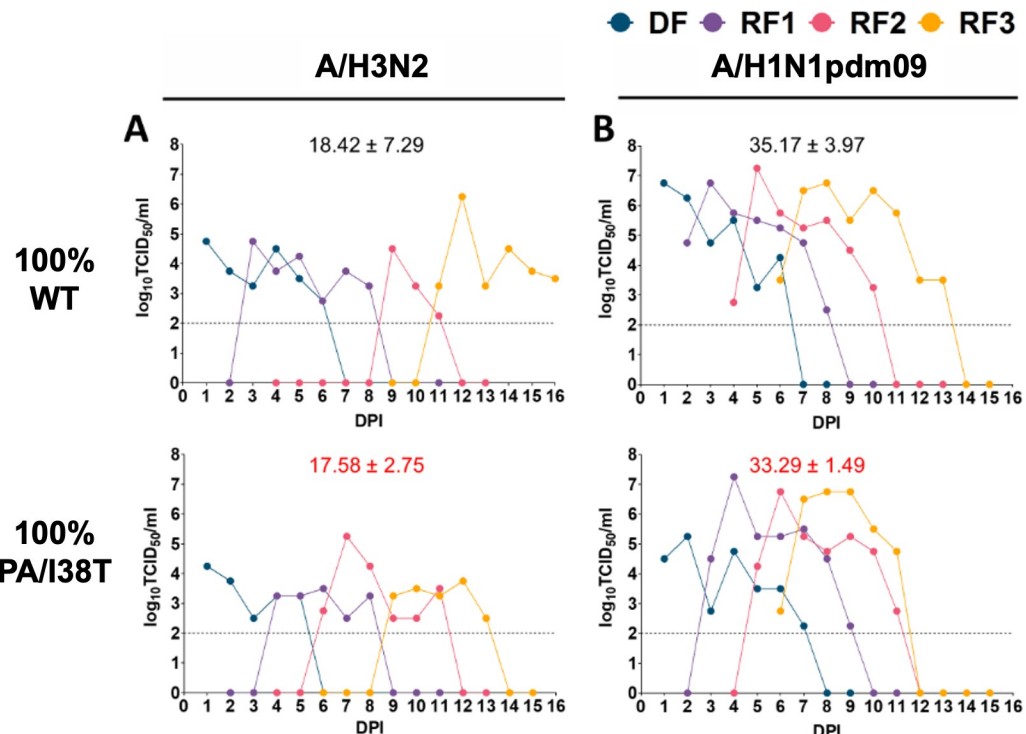

**Fig 3. Infectious viral titers (TCID$_{50}$) in nasal washes of direct contact recipient ferrets 1–3 (RF1-RF3) from pure (100%) WT and pure PA/I38T infection groups. (A) A/H3N2 clinical isolates; (B) A/H1N1pdm09 clinical isolates.**
Virus replication kinetics (TCID$_{50}$) in ferrets infected by direct contact transmission (DF and RF1-RF3) using pure populations of WT or PA/I38T-variant virus for (A) A/H3N2 or (B) A/H1N1pdm09 subtypes. Points represent infectious titers (log$_{10}$ TCID$_{50}$/mL) in individual ferret nasal washes from each DPI of DF. The mean AUC of infectious virus shedding (AUC TCID$_{50}$ ± SD) for RF1-RF3 in each group is labelled (black text, WT; red text, PA/I38T). AUC, area under the curve; DF, donor ferret; DPI, days post inoculation; RF, recipient ferret; SD, standard deviation; TCID$_{50}$, 50% tissue culture infectious dose; WT, wild type.

-18.2% (50:50), -42.3% (50:50), and -33.9% (20:80) (Fig 4). The study of within host fitness of A/H3N2 PA/I38T viruses was limited when the viral populations established in RF2-RF3 were nearly pure (Fig 4), and therefore this resulted in limited competition of WT and PA/I38T virus. The relative changes in A/H3N2 PA/I38T proportion that occurred in each animal is summarized in Fig 5A; overall there was a modest reduction in the within-host fitness of the A/H3N2 PA/I38T variant. Transmission or between-host fitness was evaluated according to the change in variant virus population in the first pyrosequencing detection (post infection) in a RF nasal wash, compared to the population detected in the cognate DF on the previous day. Between-host fitness was similar in the PA/I38T-variant and the WT A/H3N2 virus at each direct contact transmission event, with only minor changes occurring in PA/I38T proportions between DF and RF1 of -5.4% (initial proportion 80:20 [WT:I38T]), -17.8% and +1.6% (initial proportions 50:50) and +9.3% (initial proportion 20:80), with similar minor changes occurring between RF1 and RF2 transmissions (Fig 4B). Although these shifts (both increases and decreases) in %I38T were observed in the first generation of transmission (purple points), apparent transmission bottlenecks of near pure WT (left extreme of x-axis) or PA/I38T (right extreme of x-axis) had developed in the second (pink points) and third (yellow points) generations such that competition between PA/I38T and WT virus populations was effectively absent in RF2-RF3, and minimal changes in proportion were observed (Fig 5B).

## Clinical A/H3N2

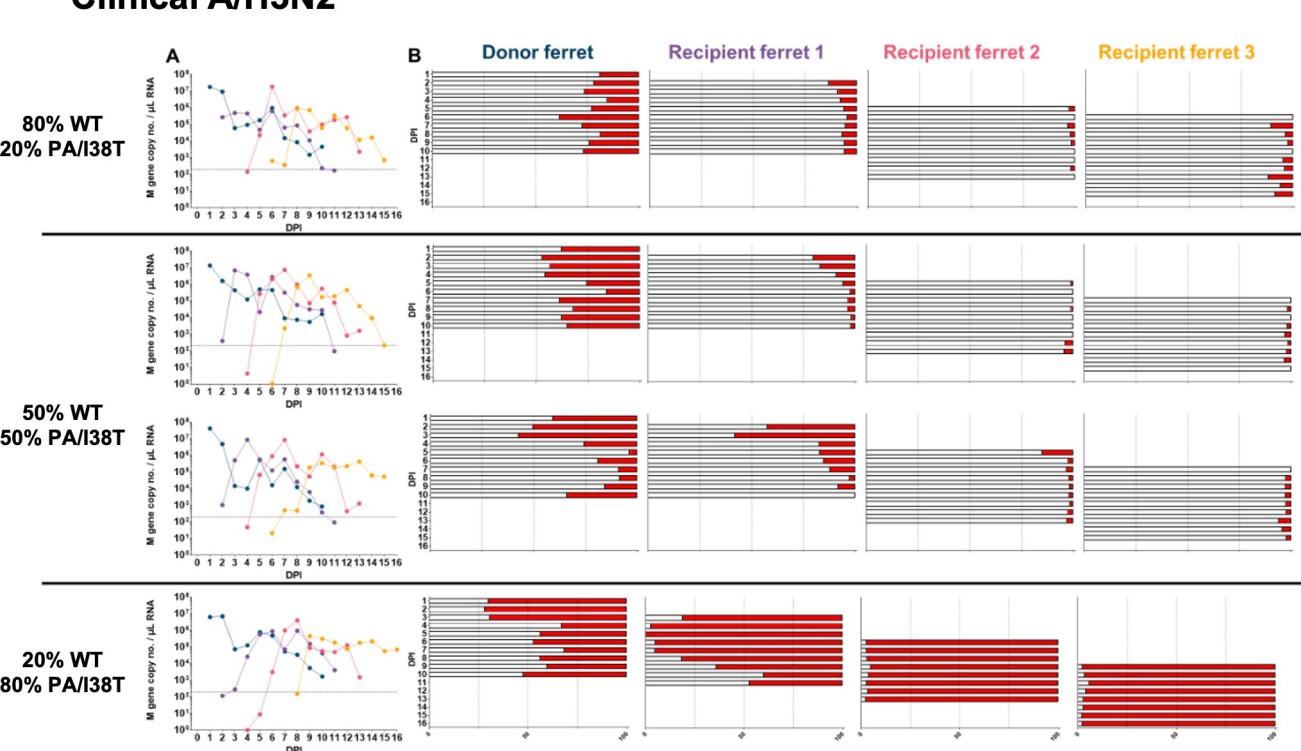

**Fig 4. A) qPCR curves for A/H3N2 viral RNA and B) WT%:I38T% of viruses in ferret nasal washes.** A/H3N2 DFs were infected (intranasal inoculation) with virus mixtures of WT and PA/I38T at a total infectious dose of $10^5$ TCID$_{50}$/ 500 µL. Daily nasal washes were collected from all animals for 10 consecutive days following inoculation/exposure, or until endpoint. RF1 was exposed to the DF by cohousing at 1 DPI. Naïve recipients were cohoused sequentially with the previous recipient (1:1) 1 day following detection of influenza A in RF nasal wash (Ct <35). (A) Viral RNA was quantified by number of M gene copies (influenza A)/µL of total RNA. (B) The relative proportion of virus encoding WT PA (white bars) and PA/I38T (red bars) in each nasal wash sample was determined by pyrosequencing. DF, donor ferret; DPI, days post inoculation; RF, recipient ferret; TCID$_{50}$, 50% tissue culture infectious dose; WT, wild type.

Overall, the populations of A/H3N2 PA/I38T-variant diminished such that they were essentially lost by RF2 in the three transmission chains that were initiated with either 20% or 50% of the variant. However, in the transmission chain initiated with 80% A/H3N2 PA/I38T, the WT virus was outcompeted, likely to be due to the impact of a small transmission bottleneck in RF1 when the variant virus was still dominant in the viral mixture at 5 DPI (Fig 4B). Therefore, across the two measured components of viral fitness, we identified that the A/H3N2 PA/I38T virus had a modest reduction in within-host fitness, but comparable between-host fitness to that of the WT virus.

**A/H1N1pdm09 PA/I38T fitness.** As observed in the A/H3N2 clinical isolate pair, infection with a pure population of the A/H1N1pmd09 PA/I38T virus was capable of three successive direct contact transmissions in the ferret model (Fig 3B), and the PA/I38T substitution was maintained without reversion to WT (S6 Fig). Similar to the *in vitro* data for 2HB001 (Fig 1A), we measured a transient reduction in mean viral titer of A/H1N1pdm09 PA/I38T recipients compared to WT at 24 hours post-exposure (p≤0.05). However, the overall infectious virus kinetics in ferrets that were infected with the A/H1N1pdm09 PA/I38T virus (n = 3) (AUC 33.29 ± 1.49) was not significantly different to those infected with the WT virus (35.17 ± 3.97, non-significant) (Fig 3B).

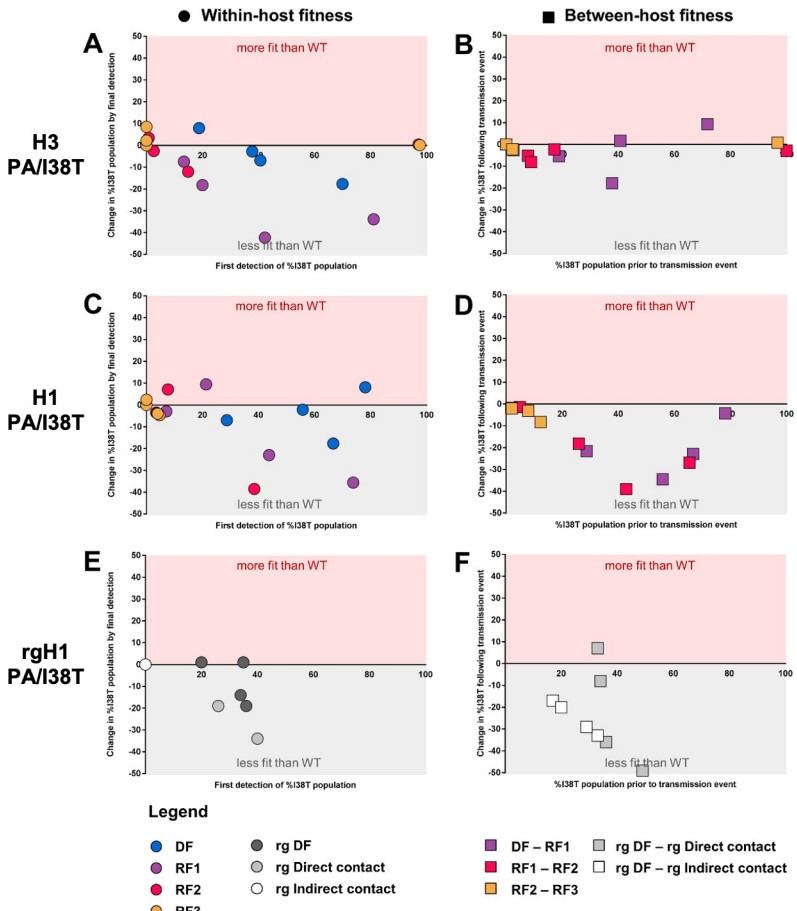

**Fig 5. Visualisations of estimated within-host fitness and between-host fitness for PA/I38T-variant viruses.**
Within-host, and between-host transmission fitness visualisation for PA/I38T-substituted clinical isolates (A–D) and recombinant viruses (E and F). (A, C and E) Within-host fitness graph Y-axis plots the change in %I38T population from first valid pyrosequencing/NGS detection (on X-axis) compared to last detection in each individual ferret (coloured circles represent DF: blue; RF1: purple; RF2: pink; RF3: yellow; rg DF: charcoal; rg direct contact: grey; rg indirect contact: white). Points in red shaded area indicate %I38T increased over time, grey shaded area indicates % I38T reduced over time. (B, D and F) Between-host fitness graph Y-axis plots the change in %I38T population for each transmission event measured using the first valid pyrosequencing/NGS measurement of %I38T in the recipient compared to the %I38T in the respective donor on the previous day (donor value is the X-axis). Coloured squares represent generations of transmission (1st, DF-RF1: purple; 2nd, RF1-RF2: pink; 3rd, RF2-RF3: yellow) or route of transmission (grey: direct contact; white: indirect contact). Points in red shaded area indicate %I38T increased following transmission, grey shaded area indicates %I38T reduced following transmission. DF, donor ferret, NGS, next generation sequencing; RF, recipient ferret.

In transmission chains initiated with mixtures of A/H1N1pdm09 WT and PA/I38T-variant viruses (Fig 2B), competitive co-infections were established at 1 DPI in DFs at similar proportions to the infectious virus ratio in the inoculum (Fig 6). In terms of within-host fitness, the proportion of A/H1N1pdm09 PA/I38T remained relatively stable over time in DFs, with changes in PA/I38T over time of -7% (initial proportion 80:20 [WT:I38T]), -2.2% and -17.7% (initial proportions 50:50) and +8.1% (initial proportion 20:80) (Fig 6). In RFs, PA/I38T also remained relatively stable overall, but with some reductions observed in the WT:I38T 20:80 chain of -35.6% (RF1) and -38.5% (RF2) (Fig 6). The overall absence of within-host fitness change in PA/I38T compared with WT is further reflected in the summary figure where the majority of within-host measurements show little or no change in viral proportion, but in a

## Clinical A/H1N1pdm09

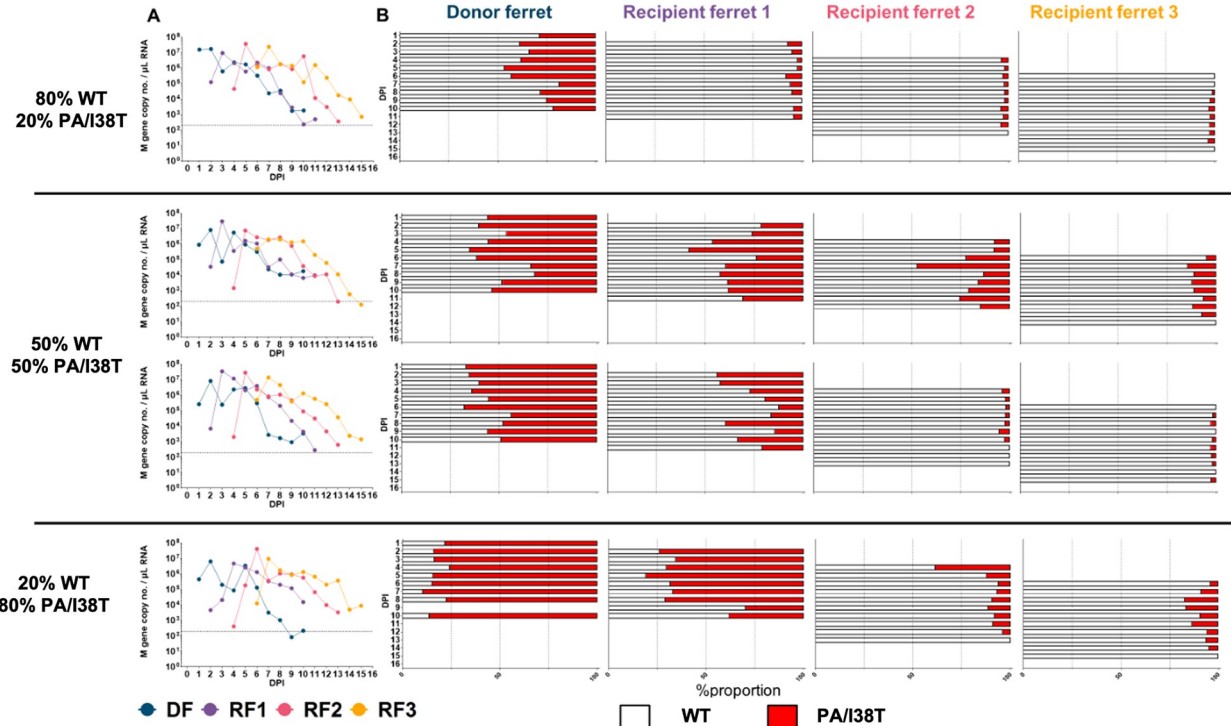

**Fig 6. qPCR curves for A/H1N1pdm09 viral RNA and B) WT%:I38T% of viruses in ferret nasal washes.** A/H1N1pdm09 DFs were infected (intranasal inoculation) with virus mixtures of WT and PA/I38T at a total infectious dose of $10^4$ TCID$_{50}$ / 500 μL. Daily nasal washes were collected from all animals for 10 consecutive days following inoculation/exposure, or until endpoint. RF1 was exposed to the DF by cohousing at 1 DPI. Naïve recipients were cohoused sequentially with the previous recipient (1:1) 1 day following detection of influenza A in RF nasal wash (Ct <35). (A) Viral RNA was quantified by number of M gene copies (influenza A) per μL of total RNA. (B) The relative proportion of virus encoding WT PA (white bars) and PA/I38T (red bars) in each nasal wash sample was determined by pyrosequencing. DF, donor ferret; DPI, days post inoculation; PA, polymerase acid; RF, recipient ferret; TCID$_{50}$, 50% tissue culture infectious dose; WT, wild type.

small number of ferrets we observed a loss of the PA/I38T-variant over time (Fig 5C). The PA/I38T-variant population decreased following every direct contact transmission event, indicating the between-host fitness of A/H1N1pdm09 PA/I38T was compromised compared to WT (Fig 5D). This observation was strongest between DF and RF1 animals (Fig 5D, purple points), where the reduction was -21.6% (initial proportion 80:20 [WT:I38T]), -34.5% and -22.9% (initial proportions 50:50) and -4.3% (initial proportion 20:80). Reductions in %I38T were also observed between RF1 and RF2 pairs (Fig 5D, pink points), and by the endpoint, the PA/I38T-variant proportion had become a minority proportion in all transmission chains (Fig 6B). Therefore, across the two measured components of viral fitness, we identified that the A/H1N1 PA/I38T virus had broadly similar within-host fitness, but substantially lower transmission fitness compared to the WT virus.

### Replication and direct/indirect contact transmission of WT/I38T reverse genetics A/H1N1pdm09 viruses in competitive fitness ferret model (London)

In the London laboratory, we tested the fitness impact of the PA/I38T amino acid substitution in a reverse genetics-derived A/H1N1pdm09 virus using both direct and indirect transmission

## Recombinant A/H1N1pdm09

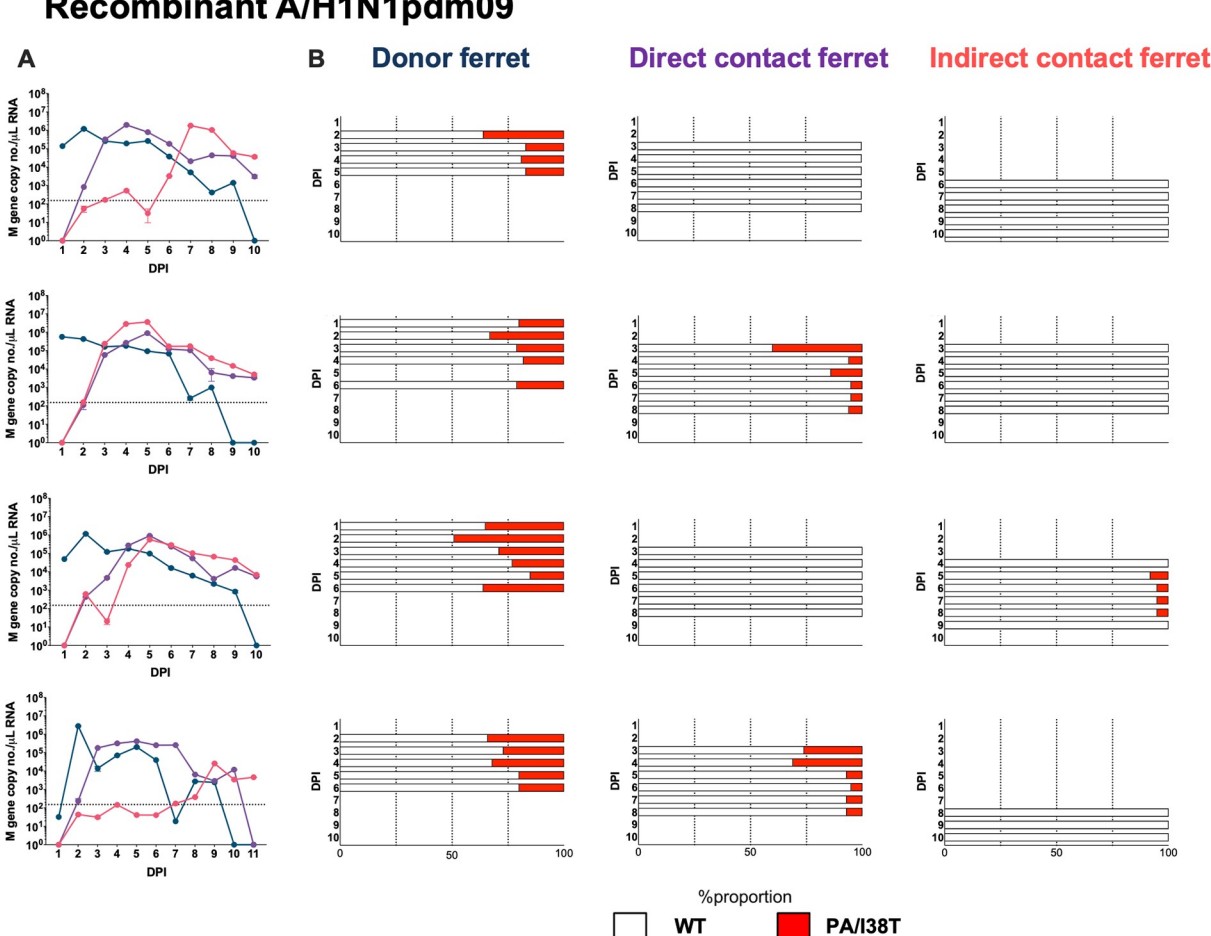

**Fig 7. Change in WT/I38T virus proportion in nasal washes of donors compared to DC or IC recipients.** Donor ferrets were infected (intranasal inoculation) with a recombinant A/H1N1pdm09 virus pair using 50:50 WT%: I38T% mixture based on infectious virus titer, at a total infectious dose of $10^4$ PFU. Naïve recipient ferrets were exposed to infected donors at 1 DPI by cohousing in the same cage (direct contact), or by housing in an adjacent cage separated by a respiratory droplet-permeable interface (indirect contact) (1:1:1 ratio). The relative proportion of virus encoding WT PA (white bars) and PA/I38T (red bars) in each nasal wash sample was determined by next generation sequencing. DPI, days post inoculation; inoc, inoculation; PA, polymerase acid; WT, wild type.

routes (Fig 2C). In the direct transmission scenario (as used in the Melbourne experiments), the naïve ferret was cohoused with the experimentally infected DF in the same cage, allowing potential transmission to occur by physical contact, aerosols/respiratory droplets, and fomites. In the indirect transmission scenario, the naïve ferret was placed in an adjacent cage separated from the DF by a perforated barrier, restricting the potential transmission route to aerosols/respiratory droplets alone. DFs were inoculated with a 50:50 WT:I38T mixture before exposure (1:1:1) to both a direct contact recipient and an indirect contact recipient. NGS was used to determine the ratio of WT:I38T virus populations in ferret nasal washes and showed that mixed infections were established in DFs at 1 DPI (Fig 7).

In terms of within-host fitness, we observed that the PA/I38T-variant population was maintained at a similar proportion in DF throughout the course of infection, but diminished rapidly in the direct contact ferrets where a mixed infection had been established by transmission (Figs 7 and 5E). In terms of between-host fitness the PA/I38T-variant virus did not transmit by direct contact in two of four occasions (Fig 7), but the proportion remained relatively stable

where transmission did occur (Fig 5F). Among indirect contact recipients, the PA/I38T-variant virus failed to transmit in three of four occasions, and in the one of four recipients where it was detected it was at a very low abundance (<10%, near limit of detection) (Fig 7). These data support the findings from the A/H1N1pdm09 clinical isolates derived from treated patients, indicating that between-host fitness is substantially compromised by the PA/I38T substitution. Furthermore, the results suggest that the transmission fitness of the PA/I38T virus is lower in the indirect transmission model compared to the direct contact model (Fig 5F).

### Detection of secondary mutations in PA/I38T variant-dominated ferrets

There is the potential that secondary mutations may have emerged in PA/I38T variant populations which may have offset fitness deficits and enabled the fixation of variant-dominated virus populations observed in the Melbourne ferret experiments. To address this, the pre-treatment WT and post-treatment PA/I38T clinical isolates used for the preparation of ferret inoculations in A/H3N2 and A/H1N1pdm09 experiments were subjected to whole genome sequencing. We then sequenced the nasal wash samples at the peak of viral shedding (by copy number) in RF3 ferrets where the PA/I38T-variant was dominant (>95% population by pyrosequencing) following three direct contact transmissions. We compared the RF3 consensus sequence to the pure PA/I38T inoculum to identify any potential compensatory mutations which may have emerged during the transmission chain.

The PA/I38T variant remained dominant in RF3 in the pure PA/I38T transmission chains in both A/H3N2 and A/H1N1pdm09 experiments (S5B and S6B Figs), but also increased in proportion in the 20% WT: 80% PA/I38T A/H3N2 group (Fig 4B). Among these RF3 ferrets, we did not detect any non-synonymous mutations which reached fixation alongside PA/I38T (S1 Table). While some subpopulations of non-synonymous mutations were detected in the RF3 ferrets, their significance is unknown (PB2/V89M amino acid substitution [at 68% of the viral population] in the pure PA/I38T transmission chain, and HA/T317I substitution [at 51% of the viral population] in 20% WT: 80% PA/I38T group) (S1 Table).

In the London experiments, the deep sequencing analysis used to quantitate PA/I38T-variant populations in ferret nasal washes identified subpopulations of non-synonymous secondary site mutants. Significant subpopulations included a PB2/R508G variant in a donor ferret (67%, 6 DPI, S1 Table), and a PB2/S36P variant (22%, 4 DPI, S1 Table) in a DC sentinel for a separate transmission group. However, these mutations occurred in WT-dominant virus populations. The recombinant PA/I38T-variant remained a minority population in all donor animals, and the recombinant WT virus became fixed in all direct/indirect contact ferrets (Fig 7B), suggesting these secondary mutations have no compensatory fitness function.

## Discussion

The detection of variant influenza viruses carrying the PA/I38T amino acid substitution in baloxavir-treated patients raises the concern that such viruses with reduced drug susceptibility could achieve sustainable transmission and circulate in the community, thereby potentially compromising the effectiveness of baloxavir usage. The present study provides evidence that the PA/I38T substitution has some impact on aspects of the intrinsic viral fitness of both A/H3N2 and A/H1N1pdm09 viruses, but that these fitness costs are modest and are not completely deleterious to viral replication and transmission.

In existing studies investigating the fitness of PA/I38T-substituted influenza viruses in animal models, variant influenza viruses of recombinant and patient-derived origins have been capable of robust viral replication in mice, hamsters and ferrets [25–28]. Ferrets represent the gold standard transmission model for evaluating the fitness of influenza viruses [29]; of the

available reports, patient-derived and recombinant PA/I38T viruses have previously been shown to transmit from experimentally inoculated DFs to naïve recipients by direct [27] and indirect contact [25,27] routes. Indirect contact models provide data exclusively on respiratory droplet/aerosol transmission, unlike direct contact models which allow for transmission by physical contact and fomites as well as airborne routes. In our study, patient-derived A/H3N2 and A/H1N1pdm09 PA/I38T-variants, when evaluated as pure viral populations, could sustain a chain of three sequential generations of direct contact transmission. However, competition experiments allowed us to determine more subtle differences in viral fitness, indicating that the treatment-emergent PA/I38T substitution compromised direct contact transmissibility of the A/H1N1pdm09 isolate, whereas the same substitution appeared to more strongly attenuate within-host replication in the A/H3N2 isolate.

Subsequently, a recombinant PA/I38T-variant A/H1N1pdm09 virus displayed similar fitness characteristics in ferrets to the patient-derived A/H1N1pdm09 isolate, and further suggested that transmission of the variant by respiratory droplets/aerosols (using an indirect contact model) was restricted to a greater extent than transmission by direct contact (Fig 7). A more restrictive transmission bottleneck exists with the indirect contact model, allowing smaller fitness differences to be more easily detected than in direct contact models. However, our current study is limited in that we did not compare the replication and transmission of pure PA/I38T infections versus WT in the indirect transmission model, and that both the donor ferret and the co-housed direct contact ferret are potential sources of infection for the naïve indirect contact in the apparatus used for the London experiment. Future studies will be needed to help determine any differences between transmissibility of WT and PA/I38T variant viruses in this regard. Our results suggest that in the absence of baloxavir selection pressure, WT influenza viruses exhibit a relative fitness advantage over PA/I38T-variants in competitive infection scenarios for both A/H1N1pdm09 and A/H3N2 subtypes; however, pure populations of PA/I38T-variant viruses were still able to be transmitted in ferrets, as also described in other studies [25,27].

Recent reports using *in vivo* competitive mixture models suggest that the fitness impact of PA/I38T can vary based on the virus background. Checkmahomed and colleagues [28] reported that PA/I38T-variants of recombinant viruses from 2009 (pandemic A/H1N1pdm09) and 2013 (seasonal A/H3N2) outcompeted cognate WT viruses within-host in mouse co-infections, whereas a separate study found a PA/I38T-substituted A/H3N2 clinical isolate from 2017 to be mildly attenuated compared with WT in ferrets [26]. Furthermore, fitness observations from competition experiments *in vitro* have not always accurately predicted *in vivo* fitness characteristics [28]. Such factors contribute to the difficulties associated with predicting the likely widespread emergence of a strain, even if the virus in question has equivalent or greater fitness than the respective WT virus being tested. A previous example of this has been observed following a competitive fitness ferret study to evaluate the fitness of an oseltamivir-resistant A/H1N1pdm09 virus with the NA/H275Y substitution that caused a localized cluster of cases among untreated patients in Australia and Japan in 2011/12 [5,6,30]. This study estimated that the oseltamivir-resistant variant virus had equivalent or even greater fitness relative to the cognate WT virus, thereby concluding that there was a significant risk that this virus would emerge and spread globally. Despite this finding in ferrets, no such dissemination of A/H1N1pdm09 NA/H275Y oseltamivir-resistant viruses in the community has occurred in the intervening years [23].

We have previously observed the impact of compensatory mutations to elevate the risk of widespread transmission for influenza viruses with reduced susceptibility to antiviral drugs. Permissive mutations in the NA and hemagglutinin (HA) proteins compensated for the intrinsic fitness costs of the NA/H275Y mutation in former seasonal A/H1N1 viruses, thereby

enabling their capacity to spread globally in 2008 [31–33]. Prior to 2007, NA/H275Y variants of former A/H1N1 seasonal viruses with reduced oseltamivir susceptibility demonstrated impaired replicative capacity *in vitro* [34]. Over time, the acquisition of several compensatory mutations resulted in the capacity for a fit NA/H275Y variant to arise and replace the seasonal A/H1N1 WT virus in 2008 [35–37]. However, with the newly emerged oseltamivir-sensitive pandemic A/H1N1pdm09 strain in 2009, the NA/H275Y A/H1N1 oseltamivir-resistant virus was subsequently outcompeted from circulation. In our present study, we utilized clinically derived and recombinant virus pairs where the PA genes differed by the PA/I38T substitution alone, demonstrating that in the absence of drug pressure, this baloxavir-resistance substitution is intrinsically detrimental to viral fitness, relative to WT. This finding is supported by the very low prevalence of influenza variants with PA amino acid substitutions known to reduce baloxavir sensitivity in global surveillance data (<0.1% in 2017–2018 through 2020–2021 [7,17,38,39]), including the 2018–19 season when over 5 million patients were treated with baloxavir in Japan [7].

However, as the PA/I38T substitution does not completely prohibit viral replication and transmission, the potential exists for additional permissive mutations to develop that recover any viral fitness loss associated with PA/I38T, which may enable increasing prevalence in community circulation. While specific compensatory mutations in PA/I38T-variants have not yet been identified, a previous report has highlighted an example of this process developing for influenza viruses with reduced susceptibility to favipiravir, another polymerase inhibitor drug. It was found that variant viruses harboring a PB1 substitution conferring resistance to favipiravir caused a loss of viral fitness and polymerase activity, which was rescued by a compensatory substitution in another part of the polymerase complex to restore the fitness and polymerase function of the variant [40]. Investigations of secondary mutations in the Melbourne ferret experiments found that none of the PA/I38T viruses contained a fixed additional substitution after three transmission events, and only two substitutions (PB2/V89M and HA/T317I) were present as mixed populations, and therefore their relevance remains unclear. No amino acid substitutions with fitness compensation potential were observed in the London experiments, as the recombinant PA/I38T-variant was consistently outcompeted by recombinant WT A/H1N1pdm09. We acknowledge that the variable sequencing protocols used by each participating laboratory limited our ability to directly compare results regarding the presence of minor population of secondary mutations. Our Sanger sequencing and pyrosequencing protocols exclusively targeted amino acids localized around position 38 of PA and may have overlooked other subpopulations that may have emerged, e.g. in ferrets prior to RF3 in the Melbourne transmission chains. Further studies using extended transmission chains and whole genome sequencing for all samples would be required to identify potentially permissive mutations for PA/I38T viruses; however, this remains a challenging subject to study prospectively.

The error-prone properties of influenza virus replication mean that substitutions leading to resistance will inevitably occur. Amino acid substitutions that confer reduced susceptibility to both the adamantanes and NAIs are well described, and likewise treatment with baloxavir has been observed to select for variant viruses that have reduced baloxavir susceptibility in clinical trials. Resistant viruses may retain sufficient fitness to transmit from person to person, typically within households [41,42]. However, from a public health perspective it is those resistant viruses that demonstrate the potential to outcompete and displace circulating antiviral sensitive strains, as happened in 2007–2009 for the oseltamivir-resistant A/H1N1 virus, that are of significant concern. Our results in the ferret model provide evidence that the intrinsic fitness of PA/I38T-substituted viruses with reduced baloxavir susceptibility is attenuated compared to baloxavir-sensitive WT strains. Indeed, despite an estimated 5.5 million doses of baloxavir supplied to Japanese medical institutions between October 2018 and January 2019 [43], there have

been very few reports of potential human-to-human spread of a PA/I38T-variant in Japanese households [43]. No clusters of sustained transmission indicating epidemiological spread of PA/I38T-variants have been identified in any population or region to date, and the number of baloxavir-resistant viruses detected in circulating viruses between September 2019 and January 2020 demonstrated that only a single PA/E23K variant in 1355 A/H1N1pdm09 viruses, and a single PA/I38M variant in 1012 A/H3N2 viruses, were detected in patients with no prior treatment [44]. Thus, it appears that baloxavir usage has not resulted in a significant number of circulating viruses with reduced susceptibility to the drug. Nevertheless, our study suggests that the fitness impact of the PA/I38T substitution in A/H1N1pdm09 and A/H3N2 influenza viruses is relatively modest, and the potential remains for the acquisition of permissive mutations to recover the fitness cost of baloxavir resistance.

## Materials and methods

### Ethics statement

The Melbourne ferret experiments were conducted with approval (AEC#1714278) from the University of Melbourne Biochemistry & Molecular Biology, Dental Science, Medicine, Microbiology & Immunology, and Surgery Animal Ethics Committee, in accordance with the NHMRC Australian code of practice for the care and use of animals for scientific purposes (8th edition). For the London ferret experiments, all work performed was approved by the local genetic manipulation (GM) safety committee of Imperial College London, St. Mary's Campus (center number GM77), and the Health and Safety Executive of the United Kingdom.

### Cells and viruses

Canine kidney MDCK and MDCK-SIAT1 cells were cultured as previously described (S1 Text) [15,16]. The human nasal epithelial cells (MucilAir) were obtained from Epithelix Sarl (Geneva, Switzerland) and the cells were maintained in MucilAir culture medium at 37°C in a humidified 5% $CO_2$ incubator.

Two A/H3N2 clinical isolate pairs were derived from human nasopharyngeal/pharyngeal swab samples from patients #344103 and #339111 from the CAPSTONE-1 phase 3 trial (otherwise-healthy patients with influenza; ClinicalTrials.gov identifier: NCT02954354) as previously described [16]. Two A/H1N1pdm09 clinical isolate pairs were derived from swabs from patients #2HB001 and #2PQ003 involved in a double-blind, multicenter, placebo-controlled phase 2 study in Japan (Japic CTI-153090) (S4 Fig). One type B clinical isolate pair was derived from a swab from patient #286102 from the CAPSTONE-1 phase 3 trial. For influenza A viruses, pre-treatment samples contained viruses without an PA/I38T substitution (WT) and post-treatment samples contained viruses with an PA/I38T substitution, and for type B viruses, WT- and PA/I38T-substituted viruses were purified from post-treatment sample by plaque pick; full-length HA, NA and PA sequences of clinical isolates from pre- and post-treatment samples were verified by Sanger sequencing (S2 Table). Clinical isolates were propagated and titrated in MDCK-SIAT1 cells. The selection criteria for clinical isolates used in the ferret study is summarized in S3 Fig.

For *in vitro* experiments, recombinant viruses based on A/WSN/33 (H1N1) and A/Victoria/3/75 (H3N2) were generated as previously described [15]. For *in vivo* experiments, a recombinant A/H1N1pdm09 virus pair of WT and PA/I38T viruses were generated by reverse genetics using plasmids containing the gene segments from A/England/195/2009 as previously described [45]. Recombinant viruses were propagated and titrated in MDCK cells.

### *In vitro* characterization of clinical isolate pairs

The baloxavir sensitivity of clinical isolate pairs (WT and PA/I38T) was assessed by plaque reduction assay in MDCK-SIAT1 cells as previously described [16]. Baloxavir acid was synthesized by Shionogi & Co., Ltd. (Osaka, Japan). The effective concentration of baloxavir achieving 50% inhibition of plaque formation ($EC_{50}$) was calculated for each virus based on plaque numbers. The mean ($\pm$ SD) $EC_{50}$ was determined from three independent experiments performed in duplicate. Data for A/H3N2 viruses (#344103 and #339111) were previously reported by Uehara et al. [16].

The replication of clinical isolate virus pairs was compared *in vitro* using primary human airway epithelial MucilAir cells (37°C, 5% $CO_2$). Confluent monolayers of MucilAir cells in transwell 24-well plates ($5.0 \times 10^5$ cells/well) were infected with either WT or PA/I38T virus at a multiplicity of infection (MOI) of 0.001 and 0.01 for type A viruses and type B viruses, respectively. The culture supernatants were collected at 0, 24, 48 and 72 hours after inoculation and the viral titer was determined by $TCID_{50}$ measured in MDCK-SIAT1 cells.

To perform competitive fitness experiments, virus pairs consisting of WT virus and the PA/I38T-substituted variant were diluted to equivalent $TCID_{50}$/mL (determined in MDCK-SIAT1 cells) and then mixed at 50:50 or 20:80 volumetric ratios. Confluent monolayers of MucilAir cells in transwell-24 plates were infected at a MOI of 0.002 and 0.02 for type A and type B viruses, respectively. Cell culture supernatant was collected 48 hours post infection and viral titers were determined by $TCID_{50}$ (S7 Fig). Fresh duplicate MucilAir cultures were inoculated with this 48 hours supernatant at an MOI of 0.001 and 0.01 for type A and type B viruses, respectively. This process was repeated for three serial passages, and the proportion of viruses harboring PA/I38X substitutions in culture supernatants at 48 hours post infection were evaluated by deep sequencing as previously described [16]. The PA coding region (nucleotides 1–466; amino acids 1–155 for A/H3N2 viruses, nucleotides 1–419; amino acids 1–139 for A/H1N1pdm09 viruses, and nucleotides 1–435; amino acids 1–145 for type B viruses) were amplified from total RNA extracted from each culture supernatant (MagNA Pure 96 system; Roche Life Science, Superscript III reverse transcriptase; Invitrogen, HotStar Taq DNA polymerase; Qiagen, Venlo, The Netherlands). Fragmented and barcoded DNA libraries were generated (Nextera XT Library Prep Kit; Illumina) and sequenced using one flow cell on an Illumina MiSeq instrument. All reads were mapped to the PA gene of the reference strains A/Texas/50/2012 (H3N2), A/California/07/2009 (H1N1) or B/Phuket/3073/2013 using the CLC Genomics Workbench Version 10.0.1. The average coverage was more than 10,000 and the percentage of reads with a quality score greater than 30 was more than 69%. A threshold frequency of >1% was adopted for calling variants, and the calculated risk of false positive variants was 0.21% for A/H3N2, 0% for A/H1N1pdm09 and type B strains, respectively, which were determined by sequencing duplicate influenza A or B virus stock samples.

### Ferret studies

**Melbourne experiments.** The replication and transmission of clinical isolate virus pairs was evaluated *in vivo* using a competitive fitness experiment design in ferrets (Fig 3B). Outbred male and female ferrets (*Mustela putorius furo*) >12 weeks old (independent vendors), weighing 600–1800 g and seronegative against recently circulating influenza viruses were used.

Virus stocks for ferret inoculation were diluted to equivalent viral titers in PBS ($10^5$ $TCID_{50}$/500 μL for the A/H3N2 viruses and $10^4$ $TCID_{50}$/500 μL for the A/H1N1pdm09 viruses) and the WT and PA/I38T-variant pairs were then mixed at various volumetric ratios: 100% WT: 0% PA/I38T, 80% WT: 20% PA/I38T, 50% WT: 50% PA/I38T (×2 groups), 20% WT: 80% PA/I38T, and 0% WT: 100% PA/I38T; n = 4 ferrets per transmission chain, total

n = 24 ferrets per experiment). Within each transmission chain, DFs were anesthetized by intramuscular injection of a 1:1 (v/v) xylazine/ketamine (10 mg/kg, Troy Laboratories) mixture, and inoculated with pure or mixed virus suspension (250 μL per nostril) by the intranasal route. The first naïve recipient (RF1) was exposed to the DF 1 DPI of virus. Ferrets were nasal washed daily under xylazine sedation (5 mg/kg, Troy Laboratories) and confirmed to be influenza positive based on a real-time reverse-transcriptase polymerase chain reaction (RT-PCR) cycle threshold ($C_t$) of <35. Following detection of virus in RF1, it was cohoused 1:1 with a subsequent naïve recipient (RF2) the next day. Similarly, once influenza was detected in RF2 it was cohoused with RF3, and the transmission chain was terminated.

Daily nasal washes were collected under xylazine sedation (0.5 mg/kg) by instilling 1 mL PBS (1% w/v BSA) into the nostrils for up to 10 consecutive days following inoculation/exposure. Nasal wash samples were stored at –80˚C until determination of infectious virus titers by in MDCK-SIAT1 cells (S8 Fig) as previously described [46]. The lower limit of quantitation (LLOQ) was 2.0 $\log_{10}$ $TCID_{50}$/mL. The experiment was terminated at 16 DPI, regardless of transmission status.

**London experiments.**   Female ferrets (20–24 weeks old) weighing 750–1000 g were used (independent vendors). After acclimatization, sera were obtained and tested by HI assay for antibodies against A/England/195/2009/H1N1pdm09. All ferrets were confirmed to be seronegative against the NP protein of influenza A virus by using an ID Screen Influenza virus A antibody competition enzyme-linked immunosorbent assay kit (ID.vet) at the start of the experiments.

DFs (n = 4) were anesthetized with ketamine (22 mg/kg) and xylazine (0.9 mg/kg) for intranasal inoculation with $10^4$ PFU of the 50% WT: 50% PA/I38T mixture of the reverse genetics A/England/195/1009 virus pair diluted in PBS (100 μL per nostril). The volumetric ratio of virus inoculation was determined based on infectious virus titer (PFU/mL). At 24 hours post inoculation, a direct contact recipient was exposed by cohousing with the donor in the same cage, while simultaneously an indirect contact recipient was placed in an adjacent cage (minimum 6 cm apart) which only allowed respiratory droplet/aerosol exposure from the donor cage (Fig 2C). Exposure of recipient animals to donor animals (1:1:1 ratio) was continued until 3 DPI (48-hour exposure period).

All animals were nasal washed daily, while conscious, by instilling 2 mL PBS into the nostrils. The nasal wash sample was used for virus titration by plaque assay in MDCK cells (S9 Fig). The LLOQ in the plaque assays was 10 PFU/mL.

## Quantitative real-time RT-PCR (qRT-PCR)

For the Melbourne experiments, viral RNA was extracted from 200 μL nasal wash samples using the NucleoMag VET isolation kit (Macherey Nagel) on the KingFisher Flex (Thermo-Fisher Scientific) platform according to manufacturer's instructions. Primer/probe sets were obtained from CDC Influenza Division: Influenza virus M gene copy number per μL RNA was determined by qRT-PCR using the SensiFAST Probe Lo-ROX One-Step qRT-PCR System Kit (Bioline) on the ABI 7500 Real Time PCR System (Applied Biosystems) under the following conditions: 45˚C for 10 min, one cycle; 95˚C for 2 min, one cycle; 95˚C for 5 s then 60˚C for 30 s, 40 cycles. Influenza A genomic copies were quantitated by the standard curve method using influenza A RNA samples of known copy number generated in-house. The M gene-targeted primers/TaqMan probe for the detection of influenza A viruses are sourced from the CDC Influenza Division (Atlanta, United States of America): forward primer (5'-ACCRATCCTGTCACCTCTGAC-3'), reverse primer (5'-GGGGCATTYTGGACAAAKCGTC-TACG-3') and TaqMan probe (6FAM-TGCAGTCCTCGCTCACTGGGCACG-BHQ1).

Results were analyzed by 7500 Fast System SDS software v1.5.1. The LLOQ was 100 M gene copies/μL RNA, based on the first RNA standard to yield $C_t$ <35.

For the London experiments, viral RNA was extracted from 140 μL nasal wash samples using the QIAamp viral RNA mini kit (Qiagen) according to manufacturer's instructions. Real-time RT-PCR was performed using 7500 Real Time PCR system (ABI) in 20 μL reactions using AgPath-ID One-Step RT-PCR Reagents 10 μL RT-PCR buffer (2X) (Thermo Fisher Scientific, Inc.), 4 μL of RNA, 0.8 μL forward (5'-GACCRATCCTGTCACCTCTGA-3') and reverse primers (5'-AGGGCATTYTGGACAAAKCGTCTA-3') and 0.4 μL probe (5'-FAM-TCGAGTCCTCGCTCACTGGGCACG-BHQ1-3'). The following conditions were used: 45˚C for 10 min, one cycle; 95˚C for 10 min, one cycle; 95˚C for 15 s then 60˚C for 1 min, 40 cycles. For each sample, the $C_t$ value for the target M gene was determined, and absolute M gene copy numbers were calculated based on the standard curve method. The LLOQ was 153 M gene copies/μL RNA, based on the results from the samples of uninfected ferrets (mean + 2*SD).

## Quantitation of PA/I38T-variant population for ferret studies

For the Melbourne experiments, detection of the PA/I38T-variant in a virus sample was performed by pyrosequencing as previously described [47]. The MyTaq One-Step RT-PCR Kit (Bioline) was used to generate and amplify cDNA from viral RNA isolated from nasal wash samples (as above). Samples containing fewer than 200 M gene copies/μL RNA were excluded from analysis. RT-PCR product was processed in the PyroMark Vacuum Workstation according to manufacturer's protocol, and pyrosequencing was performed using the PyroMark ID System (Biotage). Primers were designed to investigate SNPs at position 38 of the PA gene of influenza A/H3N2 and A/H1N1pdm09 (S1 Text). Analysis was performed using PyroMarkQ96 ID Software to quantify the percentage of WT and variant in a mixed virus population. The error associated with estimating pure populations of WT or PA/I38T virus was such that values <5% or >95% could not be accurately quantified–therefore, values <5% or >95% were considered pure WT or PA/I38T, respectively [47]. Whole-genome NGS was performed on the ferret inoculum material, and the nasal wash from the peak of viral shedding (as determined by qRT-PCR) for 100% WT RF3s, and any RF3 where the PA/I38T population reached fixation (>90% PA/I38T detected by pyrosequencing). Total viral RNA was extracted from ferret nasal washes using the QIAamp Viral RNA mini kit (Qiagen, Germany). The full genome of the viral RNA was amplified using qScript XLT 1-Step RT-PCR Kit (Quant Bio) using primers and probes previously described [48]. Library preparation and sequencing of amplicons were done utilising in-house WHOCCRRI, Melbourne protocols and on the Illumina iSeq 100 platform (2x 150 PE reads). The consensus for the inoculum samples was built utilising the CDC IRMA pipeline [49] and was used as a reference for sequences from the ferret nasal wash samples. For ferret nasal wash samples, fastq reads were also mapped to the influenza genome using Bowtie2 v2.2.5 (-very-sensitive-local) (http://bowtiebio.sourceforge.net/index.shtml). SAM tools v1.7 was used to process sequence alignments and generate pileup files. The pileup files were then used to scan for minorities using Varscan [50] with a minimum variant calling threshold set at 1%.

For the London experiments, whole-genome NGS was performed using a pipeline at Public Health England (PHE). Viral RNA was extracted from viral lysate using easyMAG (bioMérieux). One-step RT-PCR was performed using SuperScript III (Invitrogen), Platinum Taq HiFi polymerase (Thermo Fisher Scientific, Inc.), and influenza-specific primers [48]. The entire ORF of the PA segment was amplified in two overlapping fragments using two pairs of primers (S1 Text). Samples were prepared using the Nextera XT library preparation kit (Illumina) and sequenced on an Illumina MiSeq generating a 150-bp paired-end reads. Reads were mapped

with BWA (version 0.7.5) and converted to BAM files using SAMTools (version 1.1.2). Variants were called using QuasiBAM, an in-house script at PHE. Analysis of prepared minority variant mixture controls has shown that a cut-off set at 5% coupled with a minimum depth of 2000 were reliable parameters to quantify the PA/I38T-variant with 99% confidence.

We considered the PA/I38T-variant proportion measured at experiment endpoint as dependent on both within-host and between-host components of viral fitness. Within-host fitness was regarded as the observed trend in the proportion of %I38T changing in each individual ferret over time. Between-host fitness was regarded as the observed differences in the measured proportion of PA/I38T-variant in the RF compared to the proportion measured in the DF on the previous day for each transmission event.

## Statistical analysis

For *in vitro* experiments, comparisons of virus titers between WT and PA/I38T viruses at each time-point were conducted using Welch's *t*-test by using the statistical analysis software, SAS version 9.2 at the 0.05 significance level using two-sided tests. No adjustment for multiple testing was performed for exploratory analysis.

For the Melbourne *in vivo* experiments, 2-way ANOVA followed by Bonferroni's post-tests (Prism 5) were performed on daily virus titer data from pure WT- and PA/I38T-group ferrets which were infected by direct contact transmission (n = 3 per group). Data from DF were not included in this analysis as the replication kinetics of experimental infection by direct inoculation of virus differs from the surrogate for 'natural' infection that is achieved by direct contact. AUC analysis was performed, and AUC values for WT and PA/I38T groups were compared by two-tailed Mann-Whitney U test (Prism 5). $p \leq 0.05$ was considered statistically significant.

## Supporting information

**S1 Fig. Competitive replicative capacity of influenza reverse genetics-derived viruses with PA/I38 substitutions.** MDCK cells were co-infected with reverse genetics-derived WT and PA/I38-substitutited viruses at 50:50 ratio based on viral titers at an MOI of 0.001 each. At 48 hours post infection, the culture supernatant was collected and serially passaged three times. The virus passage experiments were conducted in duplicate as lineage 1 and lineage 2. Each culture supernatant was subjected to Sanger sequencing to analyze the change of amino acid at 38 in PA subunit. Sanger sequence chromatograms of amino acid at position 38 in the PA subunit were analyzed using BioPython, and blue chromatograph represents WT and red shows PA/I38-substituted viruses. Representative sequencing chromatographs of lineage 1 were shown because the similar results were obtained from both lineages. P0–P3, passage 0 to passage 3.
(DOCX)

**S2 Fig. Competitive replicative capacity of influenza reverse genetics-derived viruses with the PA/I38T substitution in the presence of various concentrations of baloxavir.** MDCK cells were co-infected with reverse genetics-derived WT and PA/I38T-substituted viruses at 50:50 ratios based on viral titers at a MOI of 0.001 each, in the presence of various concentrations of baloxavir. $EC_{50}$ values were 0.42 nmol/L and 1.13 nmol/L for rgA/WSN/33 (H1N1) and rgA/Victoria/3/75 (H3N2) viruses, respectively, based on plaque reduction assay results previously reported [15]. At 48 hours post infection, the culture supernatant was collected and serially passaged three times. The virus passage experiments were conducted in duplicate as lineage 1 and lineage 2. Each culture supernatant was subjected to Sanger sequencing to analyze the change of amino acid at position 38 in PA subunit. Sanger sequence chromatograms

of amino acid at position 38 in the PA subunit were analyzed using BioPython; the blue chromatograph represents WT and the red shows PA/I38T-substituted viruses. Representative sequencing chromatographs of lineage 1 were shown because similar results were obtained from both lineages. BXA, baloxavir acid; MOI, multiplicity of infection; N.T., not tested; P0–P3, passage 0 to passage 3. Low virus titer indicates where viruses were not passaged due to a low virus titer.
(DOCX)

**S3 Fig. Sample selection for fitness study *in vitro* and ferret transmission study.** Flow diagram of sample selection for fitness study *in vitro* and ferret transmission study for (A) influenza A/H1N1pdm09 and (B) influenza A/H3N2 viruses. Polymorphic amino acid substitutions in PA protein aligned with (A) influenza A/California/7/2009 (A/H1N1) and (B) influenza A/Texas/50/2012 (A/H3N2) are also shown.
(DOCX)

**S4 Fig. Influenza virus titer ($\log_{10}$ TCID$_{50}$/mL) in individual patients with PA/I38T-substituted viruses.** Time course of influenza virus titer of individual baloxavir-treated patients with PA/I38X-substituted viruses, determined by TCID$_{50}$ assay (values below LLOQ were set at 0.7 $\log_{10}$ TCID$_{50}$/mL). The black and red arrowheads indicate the sampling time-points for WT viruses (pre-baloxavir treatment for type A viruses and post-baloxavir treatment for type B virus) and PA/I38T-substituted viruses (post-baloxavir treatment), respectively. Black dotted line = LLOQ at 0.7 $\log_{10}$ TCID$_{50}$.
(DOCX)

**S5 Fig. Pyrosequencing of ferret nasal washes from WT or PA/I38T-variant A/H3N2 pure population groups.** (A) Viral RNA and (B) pyrosequencing of ferret nasal washes from A/H3N2 WT or PA/I38T pure population groups.
(DOCX)

**S6 Fig. Pyrosequencing of ferret nasal washes from WT or PA/I38T-variant A/H1N1pdm09 pure population groups.** (A) Viral RNA and (B) pyrosequencing of ferret nasal washes from A/H1N1pdm09 WT or PA/I38T pure population groups.
(DOCX)

**S7 Fig. Infectious viral titers (TCID$_{50}$) of MucilAir cell supernatant used for serial passaging in manuscript Fig 1.**
(DOCX)

**S8 Fig. Influenza virus titer ($\log_{10}$TCID$_{50}$/mL) of ferret nasal washes from competitive mixture direct contact transmission chains (Melbourne).** (A) A/H3N2 viral titers (B). A/H1N1pdm09 viral titers.
(DOCX)

**S9 Fig. Influenza virus titer (PFU/mL) of ferret nasal washes from transmission of recombinant A/H1N1pdm09 competition experiments (London) (blue: donor, red: direct contact, green: indirect contact).**
(DOCX)

**S1 Table. Overview of synonymous and non-synonymous mutations with $\geq$5% frequency in ferret nasal washes of PA/I38T-infected ferrets.**
(DOCX)

**S2 Table. Amino acid and nucleotide difference in A/H1N1pdm09 and A/H3N2 viruses isolated from baloxavir-treated patients.**
(DOCX)

**S1 Text. Supplementary methods.**
(DOCX)

# Acknowledgments

The authors thank Dr. Yi-Mo Deng and Jean Moselen for whole-genome sequencing and analysis of Melbourne laboratory samples, and Robert Webster (St. Jude Children's Research Hospital, Memphis) for generous provision of materials for reverse genetics experiments *in vitro*. The authors also thank Viroclinics Biosciences BV for NGS analyses on *in vitro* competitive mixtures experiments, and LSI Medience Corporation for conducting analyses on antiviral susceptibility. Editorial support was provided by John Bett, PhD, of Ashfield MedComms, an Ashfield Health company.

# Author Contributions

**Conceptualization:** Neil Collinson, Klaus Kuhlbusch, Barry Clinch, Steffen Wildum, Wendy S. Barclay, Aeron C. Hurt.

**Formal analysis:** Leo Y Lee, Jie Zhou.

**Funding acquisition:** Neil Collinson.

**Investigation:** Leo Y Lee, Jie Zhou, Paulina Koszalka, Rebecca Frise, Rubaiyea Farrukee, Keiko Baba, Shahjahan Miah, Monica Galiano, Takashi Hashimoto, Edin J. Mifsud.

**Methodology:** Leo Y Lee, Jie Zhou, Paulina Koszalka, Rebecca Frise, Rubaiyea Farrukee, Keiko Baba, Shahjahan Miah, Takao Shishido, Monica Galiano, Takashi Hashimoto, Shinya Omoto, Edin J. Mifsud, Aeron C. Hurt.

**Supervision:** Takao Shishido, Shinya Omoto, Takeki Uehara, Klaus Kuhlbusch, Steffen Wildum, Wendy S. Barclay, Aeron C. Hurt.

**Writing – original draft:** Leo Y Lee, Jie Zhou, Steffen Wildum, Wendy S. Barclay, Aeron C. Hurt.

**Writing – review & editing:** Leo Y Lee, Jie Zhou, Paulina Koszalka, Rebecca Frise, Rubaiyea Farrukee, Keiko Baba, Shahjahan Miah, Takao Shishido, Monica Galiano, Takashi Hashimoto, Shinya Omoto, Takeki Uehara, Edin J. Mifsud, Neil Collinson, Klaus Kuhlbusch, Barry Clinch, Steffen Wildum, Wendy S. Barclay, Aeron C. Hurt.

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
