## [Decision Letter · Decision Letter 0]

4 Dec 2020

Dear Dr Hurt,

Thank you very much for submitting your manuscript "Evaluating the fitness of PA/I38T-substituted influenza A viruses with reduced baloxavir susceptibility in a competitive mixtures ferret model" for consideration at PLOS Pathogens. As with all papers reviewed by the journal, your manuscript was reviewed by members of the editorial board and by several independent reviewers. In light of the reviews (below this email), we would like to invite the resubmission of a significantly-revised version that takes into account the reviewers' comments.

All three reviewers appreciated the attention to an important topic, rigorous experimental design and clear reporting of results. However, a number of comments indicated that the potential impact of the paper should be strengthened through the addition of data and/or experimental conditions. In particular, we ask that you give strong consideration to the inclusion of experiments with Baloxavir treatment, as suggested by two reviewers, and the addition of data clarifying the airborne transmission potential of the I38T virus on the H1N1 backbone in the absence of wild-type coinfection. We also feel it is important to include sequencing data from ferret nasal wash samples to address Reviewer 1's comment regarding the potential for second site mutations to modulate the fitness effects of PA I38T.

We cannot make any decision about publication until we have seen the revised manuscript and your response to the reviewers' comments. Your revised manuscript is also likely to be sent to reviewers for further evaluation.

Sincerely,

Anice C. Lowen

Associate Editor

PLOS Pathogens

Adolfo García-Sastre

Section Editor

PLOS Pathogens

Kasturi Haldar

Editor-in-Chief

PLOS Pathogens

orcid.org/0000-0001-5065-158X

Michael Malim

Editor-in-Chief

PLOS Pathogens

orcid.org/0000-0002-7699-2064

All three reviewers appreciated the attention to an important topic, rigorous experimental design and clear reporting of results. However, a number of comments indicated that the potential impact of the paper should be strengthened through the addition of data and/or experimental conditions. In particular, we ask that you give strong consideration to the inclusion of experiments with Baloxavir treatment, as suggested by two reviewers, and the addition of data clarifying the airborne transmission potential of the I38T virus on the H1N1 backbone in the absence of wild-type coinfection. We also feel it is important to include sequencing data from ferret nasal wash samples to address Reviewer 1's comment regarding the potential for second site mutations to modulate the fitness effects of PA I38T.

Reviewer's Responses to Questions

**Part I - Summary**

Reviewer #1: In this collaborative study, Lee and colleagues evaluate the within-host and between-host fitness of the I38T PA variant (associated with reduced susceptibility to baloxavir) in ferrets. Authors employ competitive mixtures (where both baloxavir-sensitive and I38T mutant viruses are mixed at different ratios prior to tandem in vitro infection or in vivo inoculation) and use NGS/pyrosequencing of cell culture supernatants and ferret nasal washes to measure maintenance and persistence of the I38T variant relative to wild-type levels. Authors employ several representative clinical isolates (inclusive of both 2009 H1N1 and seasonal H3N2 viruses) in a relevant human epithelial cell model, and conduct in vivo studies in two separate laboratories, which greatly improves the robustness of the conclusions drawn. The authors show that the I38T variant exhibits somewhat reduced fitness compared to baloxavir-sensitive viruses, though subtype-specific differences with regard to within-host and between-host fitness were present. There is a need to evaluate the relative fitness and transmissibility of influenza viruses with reduced susceptibility to currently available antiviral drugs, and studies such as this can provide helpful information towards understanding the relative likelihood of resistant viruses gaining traction in a susceptible population. The study is well-written, well-referenced, and logically organized, with figures that clearly present the changes in sequencing data over time. However, there are areas in the manuscript that would benefit from additional clarity, contextualization, and inclusion of additional control/viral titer data.

Reviewer #2: The authors’ have examined competitive fitness of influenza A viruses containing the PA I38T substitution, which confers reduced susceptibility to the antiviral drug baloxavir marboxil. A/H1N1 and A/H3N2 viruses with this PA substitution appear fully capable of contact transmission in ferrets, but in competition with WT, they exhibited impaired fitness that was more pronounced with the A/H1N1 subtype. The study adds to the growing literature of the impacts of viruses with reduced baloxavir susceptibility. There are several additions that could strengthen and clarify the current version of the manuscript.

Reviewer #3: Strengths:

1) Cleverly designed ferret studies to better understand the fitness of A/H3N2 and A/H1N1pdm09 viruses with the polymerase acidic I38T variant conferring reduced susceptibility to baloxavir relative to wild-type (WT) viruses.

2) State of the art technology and analyses to perform competition studies in ferrets to determine the relative transmission fitness of viruses in 1.

3) Results and conclusions are consistent with the notion that the A/H3N2 PA I38T mutant shows reduced within host fitness but similar between host fitness compared to wild type. In contrast A/H1N1pdm09 or similar RG versions of the virus show overall decreased fitness for strains carrying the PA I38T mutation.

4) Figures and graphs are very well done and easy to follow.

Weaknesses:

1) Minor: Perhaps it would be convenient to show the study designs as another panel on the main text and not as supplementary material.

2) The ms would have been much stronger by including a group of ferrets prophylactically treated with baloxavir on days 1 or 2 prior to contact to better establish the relative fitness advantage of the PA I38T mutants.

**Part II – Major Issues: Key Experiments Required for Acceptance**

Reviewer #1: (No Response)

Reviewer #2: 1) Treatment emergent viruses were used in the transmission portions of some experiments, which is a strength of this manuscript. With a single in vivo exception (20:80, A/H3N2), I38 viruses clearly out-compete T38. But these studies are done in the absence of drug pressure. If even sub-effective concentrations were used in contacts or donors, what do the authors expect to see? These experiments should be piloted first in the primary airway cells to see if the competition patterns are sustained. If actual effective concentrations are used in vitro or in vivo, the data may look quite different, and at minimum should be introduced to temper the conclusions made in Lines 333-344.

2) Baloxavir marboxil is effective against influenza A and B viruses (and C and D), as the authors’ mention in the introduction. I38X substitutions can also arise and negatively affect B virus baloxavir susceptibility. The authors have not addressed the competitive fitness hypothesis with B viruses containing I38T. At minimum, this could have been done in the primary airway cells. While the they suggest in vitro influenza antiviral data is not always recapitulated in vivo/in circulation (297-299), this is not an acceptable answer to avoid such experiments in a basic science manuscript.

3) The manuscript would benefit from more discussion on the types of transmission event (contact vs indirect contact) that may or may not influence onward transmission of sensitive vs reduced susceptibility virus – citing and alluding to figure 6. It’s unfortunate that different proportions or the more ‘fit’ A/H3N2 I38T virus was not used with the London Study technique. Do the authors think either contact/fomite or airborne driven transmission could influence persistence and transmission of I38X viruses?

Reviewer #3: There are no major issues with the ms. It is well done, cleverly designed, and very timely.

**Part III – Minor Issues: Editorial and Data Presentation Modifications**

Reviewer #1: Major Comments:

-Authors mention in the introduction (line 111) and discussion (lines 316-318) that compensatory mutations can arise in viruses bearing variants associated with antiviral resistance. However, it appears that sequencing of ferret nasal wash specimens was limited to evaluation of the I38T residue alone. Was the rest of the PA/other segments sequenced as well to rule out compensatory mutations that may have arisen during virus replication, or to confirm that gain-of-function mutations were not acquired during serial passage transmission experiments?

-For in vitro experiments presented in Figure 1, the authors state (line 391) that supernatant was titered 48 hrs p.i. prior to passage, but it’s unclear if this was for the first passage only or every serial passage, please clarify. Furthermore, it would be helpful if the authors could present as supplemental (or state in the text) viral titer data captured during the in vitro serial passage experiments – especially in the case of the H1N1 wt vs I38T viruses, it would be helpful to see if virus supernatants containing higher levels of wt compared with I38T (from expts presented in 1B and 1C) maintained higher viral titers in vitro to support the somewhat higher titers in growth curves for wt virus presented in Figure 1A.

-For London in vivo experiments, were airborne transmission experiments performed with the wild-type and I38T viruses in pure populations prior to the competitive mixture experiments presented in Figure 6? It is currently unclear to the reader if the reduced airborne transmission fitness of the I38T virus on the H1N1 backbone relative to wild-type is because the wild-type virus outcompeted the I38T virus, or if the I38T virus did not exhibit robust transmissibility even in the absence of wild-type competition. The Melbourne data is strengthened by data presented in Figures S4 and S5 showing kinetics of pure population groups; having similar control data for the 1:1:1 transmission experiments would be of benefit.

-Figures 3, 5, and 6 report M gene copy numbers from ferret NW samples only, but the methods state that samples were titered for the presence of infectious virus in both experiments. Could this information be made available to the reader in supplemental form or otherwise?

Minor comments:

-Please expand in the text (line 246) the first time ‘indirect transmission’ is used, to clarify that respiratory droplet/airborne route experiments are being conducted; the authors do specify this in the methods but it would be helpful to be explicit about this up-front, as ‘indirect transmission’ has also been used in the literature to describe transmission by environmental fomites.

-Please provide reasoning why the H1N1 subtype was chosen for 1:1:1 transmission studies; was it because the Melbourne studies found that this virus exhibited enhanced within-host stability compared with the H3N2 virus subtype, or a different reason?

-Please specify culture temperature for primary human airway epithelial MucilAir cells experiments (lines 380-1)

-Please disclose in the text why H3N2 clinical isolate 253104 was not tested in Figure 1C.

Reviewer #2: 1) Line 52 – It is unclear what ‘remains low’ actually means. If more contemporary surveillance data, stats, percentages, is available to the authors’, this should be stated here and also in the introduction/discussion.

2) Could the authors please confirm in the materials and methods (Line 388) that the viruses were first diluted to equivalent TCID50 units/mL, and then mixed in in various volumetric ratios?

3) I do not find the data and write-up for figure 4 to be particularly clear in the present version of the manuscript. From my perspective, some within-host/between-contacts statements are broadly derived from this figure, but may be more nuanced when taking into consideration specific aspects of figure 3 and 5. The authors’ may want to reconsider the portrayal and discussion of the data.

4) One unaddressed limitation is that different deep-sequencing protocols were used for each of the different centers or among the different in vitro and in vivo experiments, and lack of disclosure that variable protocols may miss minor populations that arise among the transmission events. In contrast, I’m actually less concerned with the ferret studies themselves being slightly variable among the centers – or being done at two different locations. Currently published baloxavir ferret data is largely in agreement, but given the costs and limited sample size associated with this model, the two-center approach is at least good insight, if not outwardly helpful to assess repeated measures. However, it's unclear why inter-center standardization of sample analyses wasn't applied.

5) Line 289 – this statement doesn’t appear to be true for the 20:80 (I:T) mixture of A/H3N2

Reviewer #3: (No Response)

PLOS authors have the option to publish the peer review history of their article (what does this mean?). If published, this will include your full peer review and any attached files.

Reviewer #1: No

Reviewer #2: No

Reviewer #3: **Yes: **Daniel R Perez
---

## [Decision Letter · Decision Letter 1]

1 Apr 2021

Dear Dr Hurt,

We are pleased to inform you that your manuscript 'Evaluating the fitness of PA/I38T-substituted influenza A viruses with reduced baloxavir susceptibility in a competitive mixtures ferret model' has been provisionally accepted for publication in PLOS Pathogens.

Best regards,

Anice C. Lowen

Associate Editor

PLOS Pathogens

Adolfo García-Sastre

Section Editor

PLOS Pathogens

Kasturi Haldar

Editor-in-Chief

PLOS Pathogens

orcid.org/0000-0001-5065-158X

Michael Malim

Editor-in-Chief

PLOS Pathogens

orcid.org/0000-0002-7699-2064

Reviewer Comments (if any, and for reference):

Reviewer's Responses to Questions

**Part I - Summary**

Reviewer #1: Authors have thoroughly addressed all comments raised during initial peer review, and have included new data (notably sequencing data) that collectively strengthens the study. No further comments.

Reviewer #2: The revised version of the manuscript has addressed the comments and critiques from my original review.

**Part II – Major Issues: Key Experiments Required for Acceptance**

Reviewer #1: (No Response)

Reviewer #2: (No Response)

**Part III – Minor Issues: Editorial and Data Presentation Modifications**

Reviewer #1: (No Response)

Reviewer #2: (No Response)

PLOS authors have the option to publish the peer review history of their article (what does this mean?). If published, this will include your full peer review and any attached files.

Reviewer #1: No

Reviewer #2: No

---

## [Editor Report · Acceptance letter]

4 May 2021

Dear Dr Hurt,

We are delighted to inform you that your manuscript, "Evaluating the fitness of PA/I38T-substituted influenza A viruses with reduced baloxavir susceptibility in a competitive mixtures ferret model," has been formally accepted for publication in PLOS Pathogens.

Best regards,

Kasturi Haldar

Editor-in-Chief

PLOS Pathogens

orcid.org/0000-0001-5065-158X

Michael Malim

Editor-in-Chief

PLOS Pathogens

orcid.org/0000-0002-7699-2064